# Using Silane Coupling Agent Coating on Acidic Aggregate Surfaces to Enhance the Adhesion between Asphalt and Aggregate: A Molecular Dynamics Simulation

**DOI:** 10.3390/ma13235580

**Published:** 2020-12-07

**Authors:** Gongying Ding, Xin Yu, Fuqiang Dong, Zezhong Ji, Junyan Wang

**Affiliations:** 1College of Civil and Transportation Engineering, Hohai University, Nanjing 210037, China; hhu_dgy@hhu.edu.cn (G.D.); hhuwjy19@hhu.edu.cn (J.W.); 2Jiangsu Expressway Engineering Maintenance Co., Ltd., Huai’an 223000, China; jzz1992@hhu.edu.cn

**Keywords:** acidic aggregate, silane coupling agent, asphalt mixture, asphalt-aggregate interface, molecular dynamics

## Abstract

Acidic aggregates have the merits of high strength and good abrasion resistance capacity. However, its poor adhesion with asphalt binder constrains its application in pavement construction. Among these, the granite aggregate is the typical one. Therefore, this study modified granite aggregates’ surface to improve their adhesion property with the asphalt binder. Specifically, the silane coupling agent (SCA) KH-560 was adopted to achieve the modification purpose. Subsequently, asphalt mixtures with modified and unmodified granite, basalt, and limestone were subjected to the boiling test, immersion test, and freeze-thaw splitting test to estimate the asphalt adhesion property. Moreover, a molecular dynamic simulation was employed to characterize the asphalt-aggregate interface from the molecular scale. The radius distribution function (RDF) and interaction energy were used as the primary indicators. The results showed that the SCA could efficiently improve the adhesion between asphalt and granite aggregates, comparable with the alkaline aggregates. In terms of the molecular scale, the incorporation of SCA could significantly increase the concentration distribution of asphalt molecules on the aggregate surface. Meanwhile, the interaction energy was correspondingly increased due to the considerable growth of non-bond interaction.

## 1. Introduction

Asphalt pavement is widely used because of its smooth surface, no joints, comfortable driving, low noise, economical cost, and easy maintenance. Its asphalt mixture, the mineral aggregates that account for more than 90% of the asphalt mixtures, contributes significantly to mechanical performance [1,2]. Good adhesion between asphalt and aggregate is one of the most basic conditions to ensure asphalt pavement performance. The Strategic Highway Research Program (SHRP) researches advocate that the polar composites in the asphalt adhere to the aggregate surface according to hydrogen bonding, Van der Waals interactions, or electrostatic forces [3]. Weak bonding between the aggregates and asphalt binder might induce moisture damage to the asphalt mixtures, resulting in the pavement’s early failure. Therefore, the demand for aggregates with high strength and a prior adhesion property with asphalt is increasing in pavement construction.

Based on the SiO_2_ content, the aggregate can be divided into three types [4,5]: alkaline aggregate (about less than 52%), neutral aggregate (about 52%~65%), and acidic aggregate (more than 65%). Alkaline minerals such as limestone and basalt are extensively adopted in China for their excellent adhesion to asphalt [6,7,8]. However, unsatisfied abrasion resistance is also another concern. Furthermore, the high demand for these minerals leads to a shortage of alkaline aggregate in local markets. Therefore, acidic aggregates such as granite may be an appropriate alternative to alkaline aggregates [9]. Although an acidic aggregate has better mechanical properties, its adhesion with asphalt is still below requirements. Thus, acidic aggregates’ poor adhesion should be efficiently resolved before its promotion in practical applications [10]. At present, there are two potential methodologies to handle this issue, as explained in Figure 1 [11,12,13]: One method is to select alkaline aggregates with a good adhesion with asphalt. Another one is to use anti-stripping agents or other methods to improve the adhesion of asphalt to acidic aggregates. As shown in Figure 1a, the addition of anti-stripping, high viscosity modifier, or alkaline aggregate powder can improve the moisture resistance of asphalt mixtures to some extent. Moreover, the acidic components can be reduced through multiple wash process. Although these methods do have some positive effects, they also cause some side effects that cannot be ignored. The anti-stripping agent has poor thermal stability; as a result, the anti-stripping agent would lose effectiveness in the short-term period due to its poor thermal stability [14]. Furthermore, an excessive anti-stripping agent will induce direct weak-linking between asphalt and aggregates [15]. Meanwhile, adding slaked lime could increase the adhesion of asphalt and aggregate. The Shell Bitumen Handbook describes the mechanism of adding slaked lime to improve the water stability of asphalt mixtures as follows: In hot-mix asphalt mixtures, slaked lime has always been used as an anti-stripping admixture, which is usually added by the mass of aggregates at 1%~3% and as a part of the filler [16]. However, if part of the quick lime in the slaked lime is not entirely digested, it will swell and significantly affect the asphalt mixture’s quality, causing pavement damage [17]. As for the multiple wash process, the dryness of the aggregate surface is hard to control, and the process of this method is complicated, restricting its promotion and application. Therefore, both the incorporation of modifiers and the multiple wash processes have certain limitations in elevating the adhesion between asphalt and acid aggregate.

According to previous research [14,18], the aggregate surface modification is supposed to be an efficient method to improve the adhesion between asphalt and aggregates, as shown in Figure 1b. Among these, the silane coupling agent (SCA) is the representative modification [18,19,20]. SCA is a modifier that contains both organic and inorganic functional groups. One type of functional group interacts with hydrophobic materials to change their polarity, and other groups promote chemical bonding with inorganic materials, thereby improving the adhesion at the interface of the mixture [21]. The mechanism of SCA action between aggregate and asphalt binder mainly includes four steps [20]. First, during hydrolysis of the coupling agent, the CH_3_ group transforms into Si–OH. Then, the Si–OH of silane can react with the aggregate surfaces’ hydroxyl group forming the covalent bond. Sequentially, the hydrogen bond in the interface also forms simultaneously. Finally, the coupling agent is adsorbed on the aggregate surface, forming molecular “bridges” to improve the interaction between the aggregate and asphalt. Karnati et al. [22] analyzed the effects of the silane coupling agents modified silica nanoparticles (SNPs) in asphalt and advocated that modified asphalt have superior anti-aging and low-temperature performance. Peng et al. [20] conducted a series of tests to research SCA’s effect on improving the adhesion between alkaline aggregate and asphalt binder. The results indicated that using SCA has a significant effect on improving adhesive interaction between aggregates and asphalt. In terms of research methods, static contact angle and pull-off tests are commonly used to study the adhesion between asphalt and aggregate. Yin et al. [10] studied the adhesion between aggregate minerals and asphalt binders using the PosiTest AT-A pull-off apparatus. Xiang et al. [23] studied SCA-modified basalt fibers’ effects on the interfacial adhesion between asphalt binder and basalt fiber, measured using the contact angle. However, few pieces of literature have studied the adhesion mechanism between asphalt and acidic aggregates at the molecular level.

Molecular dynamic (MD) simulation is a reliable and powerful approach to study the interface between asphalt and aggregate. The MD can be employed to study the physical and chemical properties of materials at the molecular scale and calculate related parameters such as adhesive strength [24], thermal expansion coefficient [25], density [26], cohesive energy density [27], radial distribution function [28], and interaction energy [29]. Xu et al. [3] established an atomic modeling approach to study the cohesive and adhesive performance of asphalt concrete and compared the experimental data to evaluate the model’s accuracy. Yaphary et al. [30] employed MD simulations to evaluate the salt environment’s effect on adhesion at the silica-epoxy interface. These studies above demonstrated that molecular simulations could be used to study the asphalt-aggregate interface’s performance at the atomistic level. Therefore, the adhesion mechanism between asphalt and SCA modified acidic aggregates can be featured through molecular dynamics simulation.

This study aims to use the silane coupling agent KH-560 to modify the granite surface’s properties and study the adhesion between the modified granite aggregate and asphalt binder. MD simulations were conducted to investigate the key factors of KH-560 to improve the interface adhesion between asphalt and granite molecular. For comparison, the adhesion characteristics of the asphalt-basalt system and asphalt-limestone system were also studied.

The study results help us understand SCA’s fundamental mechanism in elevating adhesion at the asphalt-aggregate interface.

## 2. Materials and Methodology

### 2.1. Raw Materials

The 60/80 penetration grade asphalt was selected as the virgin asphalt (Tongsha Nan Tong, China). Three kinds of aggregates, including limestone, basalt, and granite, were evaluated in this study. The KH-560 (γ-(2,3-glycidoxy) propyl trimethoxy silane) SCA was purchased from Shanghai Aladdin Biochemical Technology Co., Ltd. (Shanghai, China) and adopted to modify the granite aggregate. It is a commercial product named KH-560 [31]. All materials satisfied Chinese specifications.

### 2.2. Granite Aggregate Surface Treatment

The production of SCA modified aggregates includes three steps, as follows:Hydrolysis process of KH560: The mixed solution was prepared by mixing KH-560, distilled water, and ethanol at a ratio of 1:4:6 by mass for 6 h in a water bath at 60 °C. In what follows, the hydrolysis process of KH560 occurred at a normal temperature for 30 min to obtain a KH-560 hydrolysis solution.Get clean and dry granite: The granite aggregates were obtained by sieving, followed by the water wash and drying. Aggregates with a particle size from 13.2 to 19 mm were used in the boiling test, and aggregates with particle size under 13.2 mm were used to prepare asphalt mixtures.SCA modified granite surface: Granite aggregates were immersed in the prepared KH-560 hydrolysates solution for 5 min, and then the granites were placed in an air-dry oven pre-heated at 60 °C for one hour. Subsequently, the aggregates were heated at 170 °C with a curing time of one hour. Finally, the modified granite samples were cooled down at room temperature for the adhesion test.

### 2.3. Aggregate Wrapped by Asphalt

The aggregates selected for the test included basalt, limestone, granite, and granite treated with KH-560, which are recorded as BA, LA, GA, and GAK, respectively. The specific steps of granite treated with KH-560 have been introduced above. Before the boiling test, all aggregates (particle size from 13.2 to 19 mm) were kept clean and at a constant weight, noted as $M_{0}$. Then, each aggregate was put into the asphalt with a certain fluidity at 120 °C for 45 s until the asphalt was entirely coated on the aggregate’s surface. All the aggregates were then cooled at ambient temperature for 15 min. Subsequently, the asphalt-coated aggregate was obtained, and its mass recorded as $M_{1}$. Five parallel experiments were performed for all samples, and the mean value was calculated. Figure 2 shows the schematic diagram of the asphalt wrapped aggregate of this study.

### 2.4. Preparation of the Asphalt Mixture

AC-13 asphalt mixture was used in this study. The design of the gradation curves is presented in Table 1. The Marshall test calculated the optimal asphalt content to be 5.0%. The asphalt mixtures fabricated with BA, LA, GA, and GAK were named BA-AM, LA-AM, GA-AM, and GAK-AM.

### 2.5. Boiled Experiment

The asphalt coated aggregates were placed into a beaker of boiling water for 15 min. Afterward, these aggregates were taken out and cooled down at ambient temperature for 24 h. The weight of the aggregates was recorded once again, noted as $M_{2}$. The mass loss of asphalt after boiling, as indicated in Equation (1), can evaluate the adhesion between asphalt and aggregate by quantitative analysis.
(1)$$\eta =\frac{M_{2}-M_{1}}{M_{1}-M_{0}}\times 100\%$$

### 2.6. Tests for Asphalt Mixtures

All the asphalt mixtures were subjected to short-term and long-term aging according to T0734-2000 using the ‘Standard Test Methods of Bitumen and Bituminous Mixtures for Highway Engineering (JTG E20-2011) [32]. After preparing the samples, the aged and unaged asphalt mixture was conducted on the following tests: residual stability of immersed Marshall test and freeze-thaw splitting test.

The instrument for the Marshall stability test with force and displacement sensors was used in this study, and the residual stability of the Marshall specimen is defined using the following Equation [32]:(2)$$MS_{0}=\frac{MS_{1}}{MS}\times 100\%$$
where, $MS_{0}$ is the residual stability in the specimen, $MS_{1}$ is the Marshall stability of the specimen after water immersion (60 °C) for 48 h, MS is the Marshall stability of the specimen, and can be read automatically by the instrument.

In addition, the freeze-thaw splitting test refers to the standard (T0729–2011) [32], and the freeze-thaw splitting tensile strength ratio is defined by the following Equation [33]:(3)$$\mathrm{TSR}=\frac{RT_{2}}{RT_{1}}\times 100\%$$
where, $RT_{2}$ is the average splitting tensile strength value in the specimen after the freeze-thaw cycle, $RT_{1}$ is the average splitting tensile strength value without the freeze-thaw cycle.

## 3. Molecular Dynamics Simulation

### 3.1. Asphalt Model

An essential step in the MD simulation is to establish a molecular model of aggregate-asphalt, which refers to a clear definition of typical molecules and proportions, force fields, and energy minimization. In this respect, the three components asphalt model were chosen specially to represent overall compositions close to the real asphalt as referenced by Chen et al. [34], which proved acceptable [35]. In this study, the three-component model, including asphaltene, resin, and saturate, was used to build an asphalt model at a ratio of 5:27:41 [3]. Figure 3 shows the molecular structures of the asphalt model. Subsequently, the amorphous cell was employed to construct three-dimensional (3D) periodic structures of the asphalt model. After a geometry optimization process, the canonical ensemble (NVT) [3] was applied to keep the temperature constant at 298 K for 200 ps with a time step of 1 fs. The Andersen barostat and Nose–Hoover–Langevin (NHL) thermostat were employed to maintain a target temperature and pressure. In this study, interactions between the atoms are described in the COMPASS II force field.

### 3.2. Molecule Models for Aggregate-Asphalt

As for the aggregate, it consists of many chemical compounds. In addition to each aggregate’s primary components, there are dozens of traces of other substances, like CaO, Fe_2_O_3_, etc. A representative crystal can be selected as the aggregate model to simplify the model. The contents of the chemical components in granite [36], basalt [37], and limestone [38] aggregates are given in Figure 4. It can be found that silicon dioxide exists with a high percentage in both granite and basalt. Excessive silicon dioxide content would endow the aggregate with acidic performance. Therefore, it is assumed that silicon dioxide crystals can interpret the granite. Moreover, there is also much alumina in basalt; thus, the aluminum oxide was used to represent basalt to distinguish granite molecular from basalt molecular. Limestone consists primarily of calcium carbonate, which can be used to describe ideal limestone in the MD simulation.

The crystal cells of silica, aluminum oxide, and calcium carbonate were obtained from the Cambridge Structural Database, and the unit was cleaved in crystallographic directions at (0,0,1), (0,1,2), and (0,0,1), respectively. It was then followed by geometry transformation to a supercell, where a vacuum layer was added to form aggregate blocks with 3D periodic boundary conditions. After that, confined asphalt binder layers were placed over the silica, aluminum oxide, and calcium carbonate blocks. Subsequently, a vacuum of 30 Å was put above to avoid boundary influence. Eventually, the interface model of asphalt with granite, basalt, and limestone aggregate was obtained and recorded as granite-asphalt, basalt-asphalt, and limestone-asphalt, respectively.

For the SCA modified granite-asphalt model (granite/SCA-asphalt), Figure 2 fully illustrates that the SCA was hydrolyzed and a polymer film formed on the silica surface. This study assumed that the silica surface was all hydroxylated and formed Si–O–Si bonds with single layer SCA. Therefore, when the silica unit cell was cleaved in crystallographic directions at (0,0,1), and the surface was hydroxylated, all the hydroxyl groups were opened, forming a Si–O–Si bond with KH-560 [15]. Sequentially, confined asphalt binder layers like the granite-asphalt model and a layer of KH-560 was added between the silica and asphalt. The construction of the aggregate-asphalt models and final structures are shown in Figure 5.

Once the aggregate-asphalt system was obtained, geometry optimization was conducted to acquire a dynamic equilibration run of 100 ps subjected to an NPT ensemble. A further 200 ps NVT ensemble was used to keep the lowest energy with a time step of 1.0 fs to ensure that the model configuration was in an energy balance. All four systems were minimized using the conjugate gradient method, and the lattice parameters of the aggregate-asphalt model are shown in Table 2.

### 3.3. Interpretation of Simulation Results

#### 3.3.1. Radial Distribution Function

The adhesion of asphalt mainly depends on the properties of asphaltene. The asphaltene-asphaltene structures among molecules were evaluated by the radial distribution function (RDF), which described how the fraction density varies as a function of distance from a reference particle. The RDF g(r) is defined as in Equation (4)
(4)$$g\left(r\right)=\frac{1}{\rho 4\pi r^{2}dr}\frac{\sum_{t=1}^{T}\sum_{n=1}^{N}\Delta N\left(r\to r+dr\right)}{NT}$$
where, ρ is the density of the whole system, r is the distance, dr is the infinitely small distance, N is the number of particles between distance r and r+dr, ΔN is the number of atoms in the range of r to r+dr, and T is the total time of the simulation.

#### 3.3.2. Concentration Distribution of Asphalt Binder Components

Concentration profile, which is the mass density profile of an atom within evenly spaced slices, was employed to study the SCA effect on the concentration distribution of asphalt on the aggregate surface along the (0 0 1) direction. It was defined as dividing the entire system into several bins along a specific direction, and the relative concentration profile was calculated using Equation (5).
(5)$$\mathrm{Ralative}{\left(\mathrm{set}\right)}_{\mathrm{slab}}={\left(\mathrm{set}\right)}_{\mathrm{slab}}/{\left(\mathrm{set}\right)}_{\mathrm{bulk}}$$
where, (set)_slab_ = (number of atoms in slab)/(volume of slab) and (set)_bulk_ = (total number of atoms in the system)/(system volume).

#### 3.3.3. Interfacial Adhesion Energy

To study the adhesion ability between asphalt and SCA modified granite aggregate, the adhesion energy, interpreted as the work required for asphalt to separate from the aggregate surface, can be calculated according to Equation (6) [39]. As a comparison, the bond energies between asphalt and unmodified granite, basalt, and limestone were also calculated.
(6)$$E_{\mathrm{adhesion}}=-\Delta E_{\mathrm{inter}-\mathrm{agg}}=-(E_{\mathrm{total}}-\left(E_{\mathrm{asphalt}}+E_{\mathrm{aggregate}}\right))$$
where $\Delta E_{\mathrm{inter}-\mathrm{agg}}$ is the interaction energy between asphalt and aggregate, $E_{\mathrm{total}}$ represents the total potential energy of asphalt and aggregate, $E_{\mathrm{asphalt}}$ and represent the potential energies of asphalt and aggregate individually, respectively.

## 4. Results and Discussion

### 4.1. Effect of SCA on Adhesion between Granite Aggregate and Asphalt

The aggregates were subjected to a boiling test, and the mass loss rate of the asphalt-coated aggregate defined by Equation (1) is shown in Figure 6. The descending order of the mass loss rate of the four asphalt-wrapped aggregates followed GA > BA > LA > GAK. We found that the SCA significantly affects the adhesion between asphalt and granite, as the mass loss rate of asphalt reduced by around 75%. The mass-loss rate of GAK was slightly less than that of BA and LA. The result was because the increase of SCA hydrolysates progressively induced a polymer film formation on the GAK surface. The homogeneously formed film improved the adhesion between the asphalt and aggregates.

The SCA modified granite surface mainly changed the lipophilicity and hydrophilicity of the granite surface. This property can be tested by the contact angle method, Yang et al. [40] and Peng et al. [20]. These studies also indicated that the granite after SCA treatment has strong lipophilicity, which can significantly elevate the adhesion between asphalt and granite.

### 4.2. Effect of SCA on Residual Stability of Marshall Tests

Figure 7 presents the Marshall test results of different asphalt mixtures. The Marshall stability of the four asphalt mixtures all complied with the requirement (≥5.0 KN). We found that the Marshall stability of GA-AM and GAK-AM was higher than that of BA-AM and LA-AM. It was ascribed to the low aggregate stiffness in BA and LA. In addition, when the granite was not surface-treated, the residual stability of the unaged, short-term, and long-term aged asphalt mixtures was less than 50%. This was also lower than other mixtures under the same conditions. The results indicated that SCA could significantly improve the water stability of GAK-MA after treating the granite surface.

Moreover, under aging conditions, the stability of GAK-AM was higher than that of BA-AM and LA-AM, and its residual stability was greater than 80%, indicating that SCA had significantly improved the aging resistance of the granite aggregate. The test results showed that the SCA improved the adhesion between asphalt and granite and elevated the asphalt mixture’s water damage resistance. Similar results were reported by Yang et al. [40].

### 4.3. Effect of STA on Granite Mixture Freeze-Thaw Splitting Test

The freeze-thaw splitting test was conducted following the test method T0729-2000 in the “Test Procedure for Highway Engineering Asphalt and Asphalt Mixture” (JTG E20-2011) [32]. The four AC-13 grade asphalt mixtures’ freeze-thaw splitting tests are shown in Figure 8. For GA-AM, the freeze-thaw splitting tensile strength ratio (TSR) of the asphalt mixture after unaged, short-term aged, and long-term aged were all lower than 45%. Meanwhile, the TSR of GAK-AM before aging, short aging, and long-term aging were all higher than 75%, which indicated that GAK-AM met the minimum requirements of the specification (≥75%) under any conditions. Moreover, regardless of the conditions, the splitting strength of GAK-AM and GA-AM was much larger than that of BA-AM and LA-AM, which indicated that SCA could significantly improve the moisture sensitivity and low temperature of the granite asphalt mixtures. This result may be that granite had a higher stiffness than basalt and limestone, and under compression, the strength of the specimen was borne mainly by the aggregate. Besides, after aging, the splitting strength of the asphalt mixture increased, but TSR decreased. TSR of the GA-AM and LA-AM in short-term aging was slightly larger than that of the unaged one. It indicated that the GA-AM had a certain degree of improvement in water stability after short aging. The reason may be that during the short-term aging of asphalt, the asphaltene content increased rapidly and induced an increase of viscosity.

### 4.4. Intermolecular Distribution-Radial Distribution Functions (RDF)

In the colloidal structure of asphalt, the intermolecular interactions mainly existed between macromolecules and medium molecules. Simultaneously, asphaltene is the most vicious and most polar component of the SARA components, which is closely related to the adhesion of the aggregate [41]. Therefore, understanding the behavior of asphaltene is the basis to unveil the intermolecular interaction of asphalt binders.

Figure 9 shows the asphaltene-asphaltene pairs of different aggregate-asphalt models in terms of the RDF. The RDF curves of all four models were similar, which showed that the results were reproducible for molecular dynamics simulation for similar materials. The r value of the first peak can be used to evaluate the asphalt model’s molecular structure, and a higher first peak r means that the molecules are more likely to aggregate together [42]. It was found that the magnitude of the RDF curves for the granite/SCA-asphalt model presented a high peak value as compared to other models. The result indicated that the asphaltene had a strong tendency for aggregation. Moreover, with the increase of the r value, multiple peaks also exist, which indicates that the asphaltenes are more uniformly dispersed in the entire asphalt system after being local accumulation [43]. Prior research has reported that asphaltenes tend to aggregate and form micelles or clusters [3,39], which forms the fundamental of the colloidal structure of asphalt binder. Therefore, SCA modified granite aggregate was also able to change the asphalt binder’s behavior when bonded with acid aggregate. It is related to a large number of O–H bonds in modified aggregates [40].

Generally, the RDF peaks within 3.5 Å were primarily due to hydrogen and chemical bonds between other atoms, while the peaks at distances beyond 3.5 Å corresponded to the non-bond energy, like van der Waals and electrostatic interactions [44]. As shown in Figure 9, the RDF curves of all aggregate-asphalt models were mainly concentrated within 3.5 Å, indicating that the asphaltenes and aggregates primarily rely on non-bond interactions. Moreover, the granite/SCA-asphalt model had the most significant amplitude, indicating that the non-bond communication was the strongest between asphaltenes and SCA-modified granite. According to previous research, this non-bond interaction is mainly the Van der Waals force [3].

### 4.5. Distribution Characteristics of Asphalt Binder Components

The concentration profiles of asphaltene, resin, and saturate for four models were calculated and presented in Figure 10. When the equilibrium state was reached, the concentration profiles describe the molecular density or concentration along the vertical direction. Therefore, it provides information about the shape of the density profile change and the molecular distribution above the interface [45]. In the four models, the peak of asphaltene was always higher than that of resin and saturate, which indicated that the distribution of asphaltene was closest to the aggregate surface. Meanwhile, the resin and saturates peak concentrations performed different degrees of decreasing and widening. This result implied that the distributions of resin and saturates were relatively uniform. Meanwhile, the concentration of three components dropped sharply near the granite-asphalt model interface, while the rest of the curves of the three components were quite similar at the range of 30 to 60 Å. Moreover, in the granite-asphalt model, the concentration of asphaltenes was the lowest in the z-direction, indicating that the attraction between unsurfaced treatment granite and asphaltenes was not strong. Asphaltene’s peak concentration was significantly higher than resin and saturate, indicating that asphaltene was more likely adsorbed on the surface of aggregate. This result was consistent with Guo et al. [46]. For the granite/SCA-asphalt model, the concentration profiles of three components similar to the basalt-asphalt and limestone-asphalt model, and the resin was more likely adsorbed on the surface of granite aggregate modified by SCA. It is shown that the adhesion between asphalt and granite modified by SCA had dramatically improved.

Compared to the rest of the asphalt components, prior researchers have found that asphaltenes and resins show slightly greater concentrations around the interface [39,47]. In this study, each component’s concentration profile in the asphalt along the vertical was expressed as asphaltene > resin > saturate. The simulation results were consistent with this finding.

### 4.6. Effect of SCA on Adhesion of the Asphalt-Aggregate System

A negative value of interaction energy indicates the attraction between two components, while a positive value of interaction energy indicates repulsion [3]. Figure 11 shows the calculated interaction energy between asphalt and aggregate and non-bond components’ contributions (Van der Waals and electrostatic energy). We found that there exists an attraction between asphalt and aggregate, although the interaction energy between asphalt and aggregate varied depended on the combination of interface types. Since there is no chemical bond formed during the interaction, the adhesion between aggregate and asphalt was mainly the non-bond interaction without valence energy [3]. Simultaneously, the incorporation of SCA molecules on the granite surface increased the interaction energy by 120% for the granite-asphalt model. It implied that the SCA modified granite surface could significantly enhance the adhesion between asphalt and aggregate.

On the other hand, we noted that Van der Waals energy played a significant role in non-bonded interaction. The electrostatic interaction was negligible in the four models and served as a minor role in the asphalt–aggregate interaction.

## 5. Conclusions

The silane coupling agent KH-560 was used to modify the granite aggregate’s surface, and the boiling method, immersion Marshall test, and freeze-thaw split test were employed in the experiment. The MD simulations were applied to calculate the concentration profile and radial distribution functions of asphalt molecules on the surface of the SCA modified granite, and the interaction energy between SCA modified granite and asphalt was also studied. For comparison, the adhesion between asphalt and unmodified granite, basalt, and limestone was also studied. The research results and conclusions are drawn as follows:The SCA of KH-560 can significantly improve the adhesion between acid aggregate and asphalt. Through the boiling test, the adhesion of asphalt and aggregate obeys the order: GA < BA < LA < GAK.The immersion Marshall and freeze-thaw split test results showed that the asphalt mixture incorporating the SCA aggregates could significantly improve the low-temperature performance. Moreover, they had stronger water damage resistance and aging resistance.The existence of SCA changed the concentration distribution of asphalt binders along the vertical direction. The asphalt distribution on the granite surface after SCA modification was similar to the asphalt distribution on the basalt and limestone surface.Furthermore, after covering with the SCA, the interaction energy between asphalt and granite was close to limestone’s interaction energy. Moreover, the interaction energy between asphalt and SCA modified granite is still dominated by Van der Waals forces.

## Figures and Tables

**Figure 1 materials-13-05580-f001:**
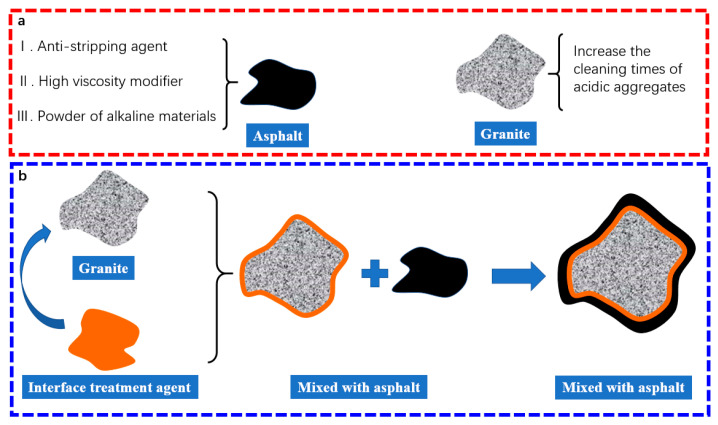
Two main methods to improve the adhesion between asphalt and aggregate: (**a**) Use modified asphalt or clean aggregate, (**b**) Modification of aggregate surface.

**Figure 2 materials-13-05580-f002:**
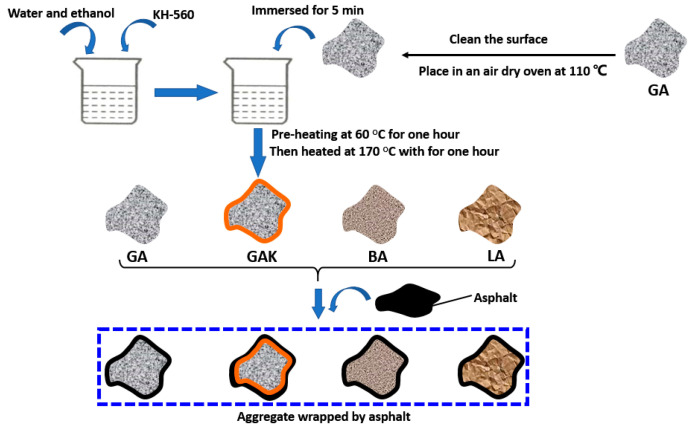
Schematic diagram of asphalt coated aggregate.

**Figure 3 materials-13-05580-f003:**
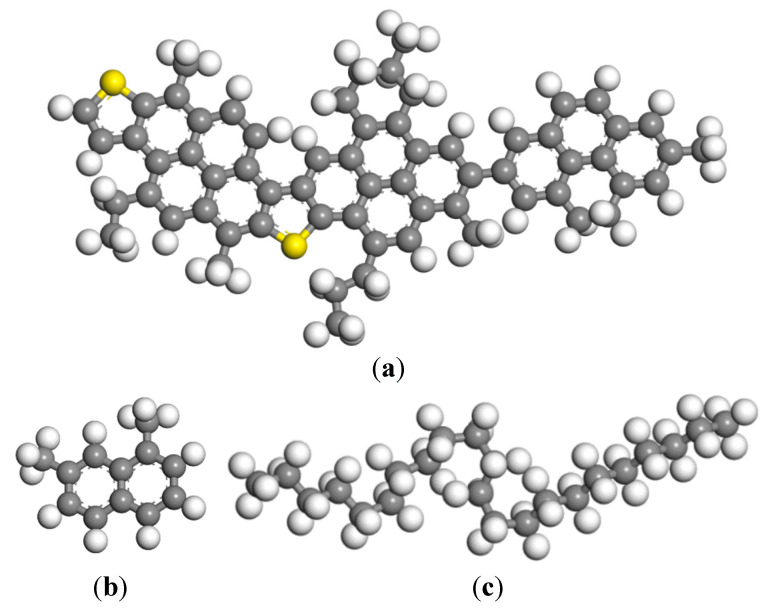
Molecular structures of (**a**) asphaltene; (**b**) resin (naphthene aromatics); (**c**) saturate (carbon atoms in gray, hydrogen atoms in white, and sulfur atoms in yellow).

**Figure 4 materials-13-05580-f004:**
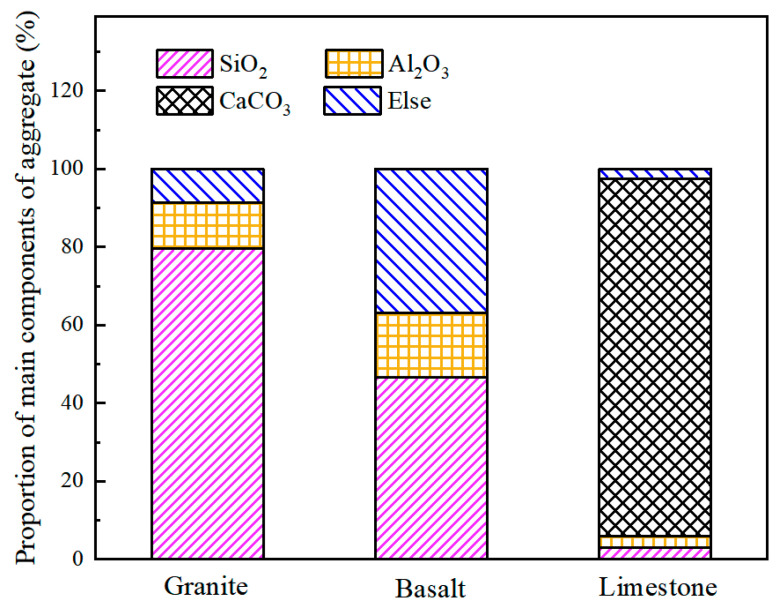
Aggregates and its major component.

**Figure 5 materials-13-05580-f005:**
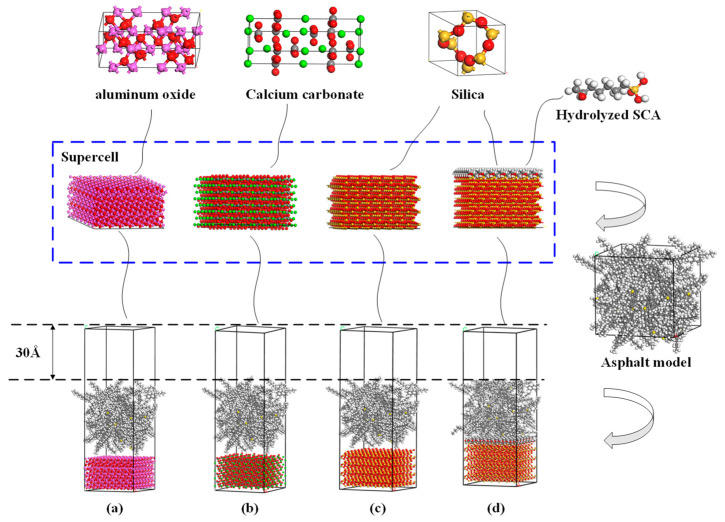
Process of building the model of (**a**) the basalt-asphalt, (**b**) the limestone-asphalt, (**c**) the granite-asphalt, and (**d**) the granite/SCA-asphalt.

**Figure 6 materials-13-05580-f006:**
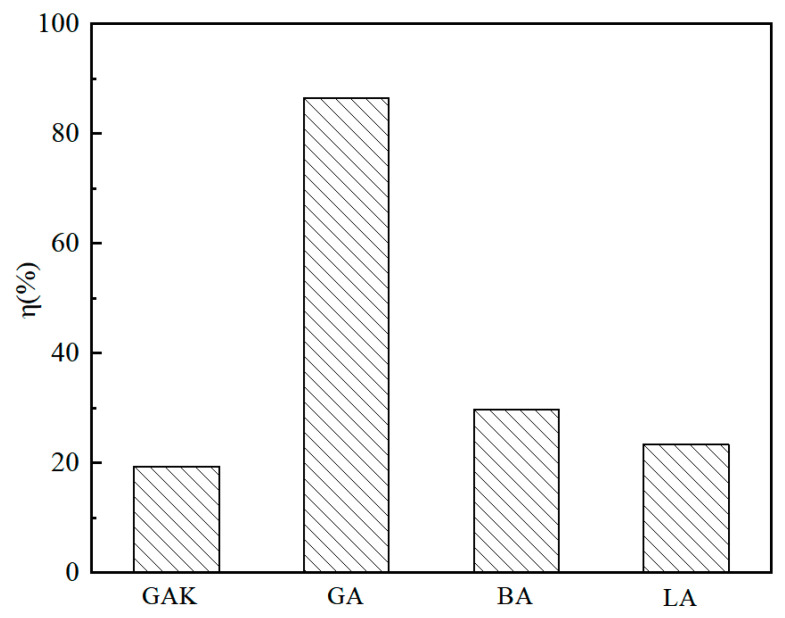
The mass-loss rate of asphalt coated different aggregates in the boiling test.

**Figure 7 materials-13-05580-f007:**
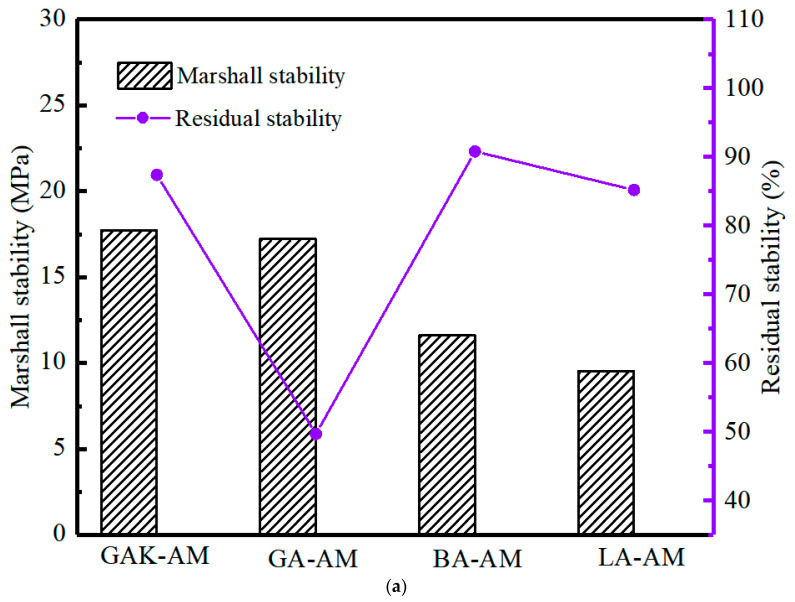
Mixture Marshall test of asphalt mixture (**a**) unaged, (**b**) short-term aged, (**c**) long-term aged.

**Figure 8 materials-13-05580-f008:**
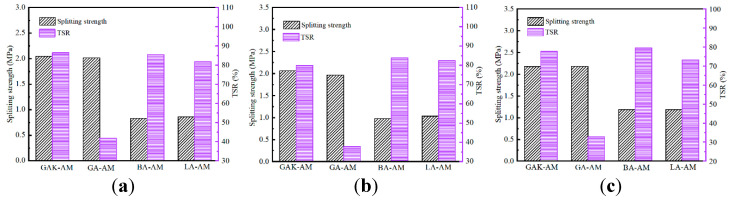
Freeze-thaw splitting tests of asphalt mixture: (**a**) unaged, (**b**) short-term aged, (**c**) long-term aged.

**Figure 9 materials-13-05580-f009:**
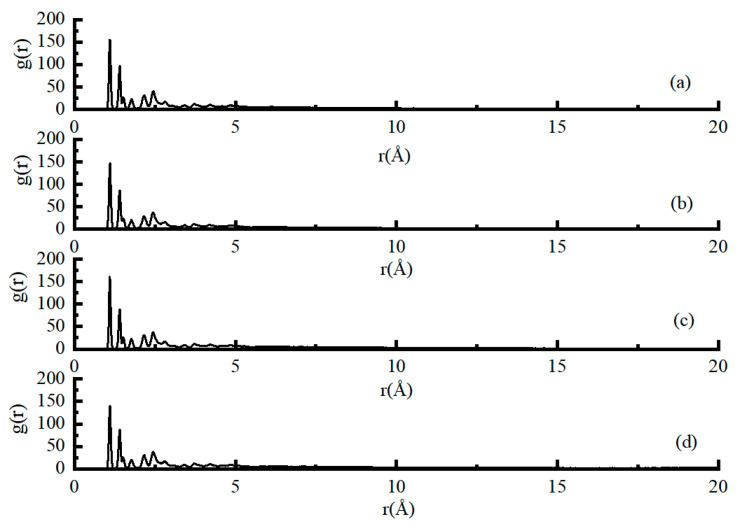
The radial distribution function for asphaltene-asphaltene, (**a**) granite/SCA-asphalt model, (**b**) granite-asphalt model, (**c**) basalt-asphalt, and (**d**) limestone-asphalt.

**Figure 10 materials-13-05580-f010:**
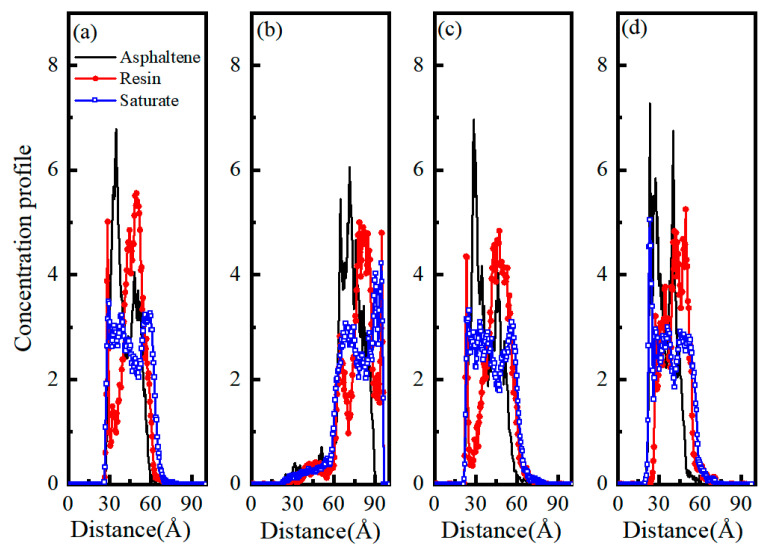
The concentration profiles for three components of asphalt along the vertical direction: (**a**) granite/SCA-asphalt model, (**b**) granite-asphalt model, (**c**) basalt-asphalt, and (**d**) limestone-asphalt.

**Figure 11 materials-13-05580-f011:**
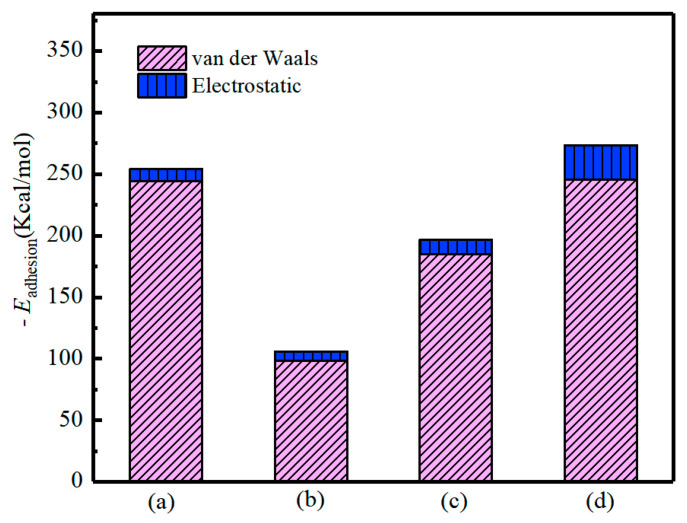
Illustration of asphalt-aggregate interaction energy and its component: (**a**) granite/SCA-asphalt model, (**b**) granite-asphalt model, (**c**) basalt-asphalt, and (**d**) limestone-asphalt.

**Table 1 materials-13-05580-t001:** Gradation of the gravel used for asphalt mixture.

Parameter		Passing Percentage (%)								
Sieve Size (mm)		13.2	9.5	4.75	2.36	1.18	0.6	0.3	0.15	0.075
Limit of gradation	Upper	100	85	68	50	38	28	20	15	8
	Lower	90	68	38	24	15	10	7	5	4
Synthetic gradation	GA/GAK	94.1	79.6	47.9	32.4	23.5	17.3	12.0	8.8	6.1
	BA	94.9	73.4	45.6	29.5	20.1	12.7	9.1	6.8	5.9
	LA	97.5	74.7	43.2	30.1	19.8	13.0	10.1	7.4	5.5

**Table 2 materials-13-05580-t002:** Lattice parameters of the aggregate-asphalt model.

Model	Lattice Parameters
granite-asphalt	*a* = *b* = 33.628 Å, *c* = 102.932 Å, α = β = γ = 90°
granite/SCA-asphalt	*a* = *b* = 33.628 Å, *c* = 90.323 Å, α = β = γ = 90°
basalt-asphalt	*a* = *b* = 33.089 Å, *c* = 97.484 Å, α = β = γ = 90°
limestone-asphalt	*a* = *b* = 32.624 Å, *c* = 103.277 Å, α = β = γ = 90°

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
