# Peer review of "Using Silane Coupling Agent Coating on Acidic Aggregate Surfaces to Enhance the Adhesion between Asphalt and Aggregate: A Molecular Dynamics Simulation"

_materials, 2020, doi:10.3390/ma13235580_

Round 1

Reviewer 1 Report

These are reviewers comments for the manuscript entitled: 'Using silane coupling agent coating on acidic aggregate surfaces to enhance the adhesion between asphalt and aggregate: A molecular dynamics simulation'. Authors have presented a very clear and comprehensive study and presented the argument behind the work (literature review), methods, and results very well. 

My only negative comments is that authors did not conclude and discuss the work. Please revise the conclusions.

Well done of work done!

Reviewer 2 Report

General Comment

This manuscript shows a very interesting experimental approach to further improve and boost the use of granite aggregates in asphalt roads construction. Although, the silylation process to enhance the coating of mineral aggregates is well known, the incorporation of the molecular dynamics simulation can help to understand the chemical mechanisms governing the surface treatment. Unfortunately, the current version of the manuscript is very poor and a deep revision must be addressed. This must be included an improvement of the introduction where the frame of the study was well defined, a better description of the experimental tests as well as a more scientific discussion of the results. Furthermore, a professional assistance with the English writing would be required. Therefore, I must suggest the rejection of this manuscript for its publication in Materials.

Next, I describe some detailed comments to help the authors to improve specific details.

Detailed Comments

  • Consider to define the alkaline aggregates as basic aggregates as for the acidic aggregates.
  • Explain better the chemical mechanisms (OH groups, polar groups,…) behind the adhesion between bituminous binder and the surface of different types of mineral aggregates.
  • Clarify why you say that the acidic aggregates can be an appropriate alternative. Do they have better abrasion resistance and/or more availability in local markets?
  • Review Figures numbers and descriptions (captions). For example, you should introduce Figure 1 before referring to it in the text. In addition, you must improve it because it has three parts (a., b. and c. Mechanism of SCA). You need to use a more general caption that describe jointly these three parts.
  • Figure 1c. Explain in detail here each step (1,2,3,4) or mention you are going to do it in section 3.2
  • Add reference to support the statement: "the excessive anti-stripping agent will induce direct weak-linking between asphalt and aggregates".
  • Specify how "slaked lime" is added. Which method? (as filler? anti-stripping agent? modifier?)
  • Figure 1b does not show the improvement of the adhesion between asphalt and aggregates but rather explain the process in a scheme. You must be more accurate.
  • I would suggest you to clarify that you used "a commercial silane couple agent known or named as KH-560".
  • Include details about the coating process in Figure 2 (e.g. asphalt temperature). Consider the use of "coated" instead of "wrapped" aggregate.
  • Clarify how much percentage of GAK was used in the asphalt mixtures and if you are using the treated aggregated described in section 3.3. It is not clear if you treated and pre-coated only mineral fractions 13.2 to 19 mm or the treatment was done for all the sizes. This is strange because 13.2 was the maximum aggregate size in AC13 mixtures.
  • Define how you obtained Marshall and residual stability (if standard process is followed, please add it).
  • Describe the freeze-thaw splitting method as well.
  • Figure 7. I highly recommend using a different type of chart. It can be misleading to include both "stabilities" together without specifying their units. It could seem that after water immersion the performance is improving.
  • Any comment on the worse performance (Marshall stability) of BA and LA in comparison to GA?
  • What TSR means?

Reviewer 3 Report

Dear Author, thank you for your manuscript. Here will be few remarks to your paper:

Line 6: ‘and’ is used between last co-author name in line.

Line 35: ‘alkaline minerals’, ‘alkaline rocks’

Line 41: in figure is mentioned – granite, and in the text only limestone and basalt.

Line 54: please add references to the research here

Lines 90-97: can be included as last paragraph of the Introduction

Chapter 3: Materials and Methodology

Author Contributions: missing info

Funding: missing info

Reviewer 4 Report

The article demonstrates valuable and interesting research results both experimental and theoretical. The manuscript concerns research on the improvement of adhesion between asphalt and granite aggregates with (SCA)  KH-560 application. The topic of the thesis has been correctly formulated and corresponds to the content, the keywords are also well suited to the discussed issues. The structure of the article is correct and logical. The presented figures and tables are correct and legible. The presented discussion is correct in terms of content and the conclusions are clearly defined.  Congratulations to the authors of the interesting research results. However, I have a few questions that need clarification:

What is the reason for choosing KH-560. Are there other similar measures on the market? What were the criteria for selecting this particular KH-560?

In my opinion, a brief information on this is required.

Another caveat is the lack of information about the properties of HK-560. There is no end to their description, in my opinion it is enough to supplement the publication with appropriate references. You can give, for example Wang, X .; Wang, L .; Ji, W .; Hao, Q .; Zhang, G .; Meng, Q. Characterization of KH-560-Modified Jute Fabric / Epoxy Laminated Composites: Surface Structure, and Thermal and Mechanical Properties. Polymers 2019, 11, 769.

Work in many areas is more of an expertise than scientific work. In the case of the discussed issue, it is beneficial for this particular work.
I congratulate the authors on their very interesting work and results and wish them good luck in the publication of this work.

Round 2

Reviewer 2 Report

I still believe that this is a good experimental work whose results could be very useful and potentially exploited. Although the authors have followed the reviewers' recommendations, I honestly think the manuscript must be improved from a writing point of view. Even after a quick look at the current version, several grammar mistakes have been found. In my opinion, a proper revision of the manuscript by a professional writing/editing service could extremely upgrade the quality of the work.